# Drug Resistance Prediction Using Deep Learning Techniques on HIV-1 Sequence Data

**DOI:** 10.3390/v12050560

**Published:** 2020-05-19

**Authors:** Margaret C. Steiner, Keylie M. Gibson, Keith A. Crandall

**Affiliations:** 1Computational Biology Institute, Milken Institute School of Public Health, The George Washington University, Washington, DC 20052, USA; kmgibson@gwu.edu (K.M.G.); kcrandall@gwu.edu (K.A.C.); 2Department of Biostatistics and Bioinformatics, Milken Institute School of Public Health, The George Washington University, Washington, DC 20052, USA

**Keywords:** HIV, antiretroviral therapy, HIV drug resistance, machine learning, deep learning, neural networks

## Abstract

The fast replication rate and lack of repair mechanisms of human immunodeficiency virus (HIV) contribute to its high mutation frequency, with some mutations resulting in the evolution of resistance to antiretroviral therapies (ART). As such, studying HIV drug resistance allows for real-time evaluation of evolutionary mechanisms. Characterizing the biological process of drug resistance is also critically important for sustained effectiveness of ART. Investigating the link between “black box” deep learning methods applied to this problem and evolutionary principles governing drug resistance has been overlooked to date. Here, we utilized publicly available HIV-1 sequence data and drug resistance assay results for 18 ART drugs to evaluate the performance of three architectures (multilayer perceptron, bidirectional recurrent neural network, and convolutional neural network) for drug resistance prediction, jointly with biological analysis. We identified convolutional neural networks as the best performing architecture and displayed a correspondence between the importance of biologically relevant features in the classifier and overall performance. Our results suggest that the high classification performance of deep learning models is indeed dependent on drug resistance mutations (DRMs). These models heavily weighted several features that are not known DRM locations, indicating the utility of model interpretability to address causal relationships in viral genotype-phenotype data.

## 1. Introduction

Human immunodeficiency virus (HIV), which causes acquired immunodeficiency syndrome (AIDS), affects over 1.1 million people in the U.S. today [1]. While there is still no cure, HIV can be treated effectively with antiretroviral therapy (ART). Consistent treatment with ART can extend the life expectancy of an HIV-positive individual to nearly as long as that of a person without HIV by reducing the viral load to below detectable levels [2] and can reduce transmission rates [3,4]. However, the fast replication rate and lack of repair mechanisms of HIV leads to a large number of mutations, many of which result in the evolution of HIV to resist antiretroviral drugs [5,6]. Drug resistance may be conferred at the time of HIV transmission, so even treatment-naïve patients may be resistant to certain ART drugs, which can lead to rapid drug failure [7]. As a result, the analysis of drug resistance is critical to treating HIV, and thus is an important focus of HIV research.

HIV drug resistance may be directly evaluated using phenotypic assays such as the PhenoSense assay [8]. Both the patient’s isolated HIV strain and a wild type reference strain are exposed to an antiretroviral drug at several concentrations, and the difference in effect between the two indicates the level of drug resistance. Resistance is measured in terms of fold change, which is defined as a ratio between the concentration of the drug necessary to inhibit replication of the patient virus and that of the wild type [9]. These tests are laborious, time-intensive, and costly. Additionally, phenotypic assays are reliant on prior knowledge of correlations between mutations and resistance to specific drugs, which evolve quickly and thus cannot be totally accounted for [10]. An alternative, the “virtual” or genotypic test, predicts the outcome of a phenotypic test based on the genotype using statistical methods. Several web tools exist for HIV drug resistance prediction based on known genotype profiles, including HIVdb [11] and WebPSSM [12], utilizing rules-based classification and position-specific scoring matrices, respectively, to predict drug resistance. Two additional tools utilize machine learning approaches: geno2pheno [13] and SHIVA [14]. 

Several machine learning architectures have been applied to predict drug resistance, including random forests such as those used in SHIVA [14,15,16], support vector machines as in geno2pheno [13], decision trees [17], logistic regression [16], and artificial neural networks [10,18,19,20]. Deep learning models (i.e., neural networks) are a major focus in current machine learning research and have been successfully applied to several classes of computational biology data [21]. Yet, one aspect of deep learning models in particular that has been overlooked to date is model interpretability. Deep learning is often criticized for its “black box” nature, as it is unclear from the model itself why a given classification was made. The resulting ambiguity in the classification model proves to be a major limitation to the utility of deep learning in translational and clinical applications [22]. As a response to this concern, recently-developed model interpretability methods seek to map model outputs back to a specific subset of the most influential inputs, or features [23]. In practice, this method allows researchers to better understand whether predictions are based on relevant patterns in the training data as opposed to bias, thus attesting to the model’s trustworthiness and, in turn, providing the potential for deep learning models to identify novel patterns in the input data.

Here, we integrate deep learning techniques with HIV genotypic and phenotypic data and analyses in order to investigate the implications of the underlying evolutionary processes of HIV-1 drug resistance for classification performance and vice versa. The objectives of this study are: (i) to compare the performance of three deep learning architectures which may be used for virtual HIV-1 drug resistance tests–multilayer perceptron (MLP), bidirectional recurrent neural network (BRNN), and convolutional neural network (CNN), (ii) to evaluate feature importance in the context of drug resistance mutations, and (iii) to explore the relationship between the molecular evolution of drug resistant strains and model performance.

## 2. Methods 

Briefly, our deep learning approach included training and cross-validation of three architectures for binary classification of labeled HIV-1 sequence data: MLP, BRNN, and CNN (Figure 1). In addition to comparing performance metrics across architectures, we evaluated feature importance using a permutation-based method and interpreted these results using known DRM loci. Lastly, we reconstructed and annotated phylogenetic trees from the same data in order to assess clustering patterns of resistant sequences.

### 2.1. Data

Genotype-phenotype data were obtained from Stanford University’s HIV Drug Resistance database, one of the largest publicly available datasets for such data [9]. The filtered genotype-phenotype datasets for eighteen protease inhibitor (PI), nucleotide reverse transcriptase inhibitor (NRTI), and non-nucleotide reverse transcriptase inhibitor (NNRTI) drugs were used (downloaded August 2018; Table 1). While the Stanford database additionally includes data pertaining to integrase inhibitors (INIs), at the time of this study insufficient data were available for deep learning analysis, and so here we have focused on polymerase. In the filtered datasets, redundant sequences from intra-patient data and sequences with mixtures at major drug resistance mutation (DRM) positions were excluded from further analyses. In total, the data pulled from the Stanford database contained 2112 sequences associated with PI susceptibility, 1772 sequences associated with NNRTI susceptibility, and 2129 sequences associated with NRTI susceptibility (Table 1). Drug susceptibility testing results included in the Stanford dataset had been generated using a PhenoSense assay [8]. Susceptibility is expressed as fold change in comparison to wild-type HIV-1; a fold change value greater than 3.5 indicates that a sample is resistant to a given drug [10,17].

All sequences were from regions of the polymerase (*pol*) gene, which is standardly sequenced in studies of HIV-1 drug resistance. Sequences in the PI dataset include a 99 amino acid sequence from the protease (*PR*) region (HXB2 coordinates: 2253-2549), while the NRTI and NNRTI datasets include a 240 amino acid sequence from reverse transcriptase (*RT*) region (HXB2 coordinates: 2550-3269). Subtype is not explicitly listed in the filtered dataset, though the consensus sequence given is that of Subtype B. The data were separated into 18 drug-specific datasets, each of which contained all sequences for which a resistance assay value for the drug was available, as well as their respective drug resistance status. For training and evaluating deep learning models, amino acid and ambiguity codes were encoded via integer encoding.

### 2.2. Deep Learning Classifiers

Three classes of deep learning classifiers were constructed, trained, and evaluated in each of the 18 datasets: multilayer perceptron (MLP), bidirectional recurrent neural network (BRNN), and convolutional neural network (CNN). The structure and parameters of each model are described below (Section 2.2.1, Section 2.2.2, Section 2.2.3). All data pre-processing, model training, and model evaluation steps were completed using R v3.6.0 [24] and RStudio v1.2.1335 [25]. All classifiers were trained and evaluated using the Keras R package v2.2.4.1 [26] as a front-end to TensorFlow v1.12.0 [27], utilizing Python v3.6.8 [28].

All classifiers were evaluated using 5-fold stratified cross-validation with adjusted class weights in order to account for small dataset size and class imbalances (Table 1), implemented as follows. All data were randomly shuffled and then split evenly into five partitions such that each partition had the same proportion of resistant to non-resistant sequences. Then, a model was initiated and trained using four of the five partitions as training data and the fifth as a hold-out validation set. Next, a new, independent model was initiated and trained using a different set of four partitions for training and the fifth as the hold-out validation set. This was repeated three additional times such that each partition was used as a validation set exactly once. Thus, for each architecture and dataset, a total of five independent neural networks models were trained and evaluated using a distinct partition of the total dataset for validation and the remainder for training. This method is advantageous when working with limited available data, as it allows all data to be utilized towards evaluating the performance of a given architecture without the same data ever being used for both training and validation of any one model. Taking the average of performance metrics produced from each fold also lessens the bias that the choice of validation data may inflict on these results. 

The class weights used to train each model were calculated such that a misclassification of a non-resistant sequence had a penalty of 1 and the misclassification of a resistant sequence had a penalty of the ratio of non-resistant to resistant sequences (i.e., a misclassification of a resistant sequence had a penalty higher than that of a non-resistant sequence by a factor of the observed class imbalance). Each model was compiled using the Root Mean Square Propagation (RMSprop) optimizer function and binary cross-entropy as the loss function, as is standardly used for binary classification tasks, and was trained for 500 epochs with a batch size of 64.

#### 2.2.1. Multilayer Perceptron

A multilayer perceptron (MLP) is a feed-forward neural network consisting of input, hidden, and output layers that are densely connected. MLPs are the baseline, classical form of a neural network. The MLP used here included embedding and 1D global average pooling layers (input), followed by four feed-forward hidden layers each with either 33 (PI) or 99 (NNRTI or NRTI) units, Rectified Linear Unit (ReLU) activation, L2 regularization, and ending with an output layer with a sigmoid activation function. 

#### 2.2.2. Bidirectional Recurrent Neural Network

A bidirectional recurrent neural network (BRNN) includes a pair of hidden recurrent layers, each of which feeds information in opposite directions, thus utilizing the forward and backward directional context of the input data. BRNNs are commonly used in applications where such directionality is imperative, such as language recognition and time series data tasks. The BRNN used here included one embedding layer, one bidirectional long short-term memory (LSTM) layer with either 33 (PI) or 99 (NNRTI or NRTI) units, dropout of 0.2, and recurrent dropout of 0.2, and ended with an output layer with a sigmoid activation function.

#### 2.2.3. Convolutional Neural Network

A convolutional neural network (CNN) operates in a manner inspired by the human visual cortex and consists of convolutional (feature extraction) and pooling (dimension reduction) layers. CNNs are well known for their use in computer vision and image analysis but have recently been applied to genetic sequence data with much success, especially for training on DNA sequence data directly [21]. The CNN used here included one embedding layer, two 1D convolution layers with 32 filters, kernel size of 9, a ReLU activation function, and one 1D max pooling layer in between, then ending in an output layer using a sigmoid activation function. 

### 2.3. Performance Metrics

The following metrics were recorded for each evaluation step: true positive rate/sensitivity (TPR), true negative rate/specificity (TNR), false positive rate (FPR), false negative rate (FNR), accuracy, F-measure (F1), binary cross entropy (loss), and area-under-the-receiver operating characteristic curve (AUC). Formulas for the calculation of all metrics are given below. Binary cross-entropy was reported in Keras output and AUC and receiving operator characteristic (ROC) curves were generated using the ggplot2 v3.1.1 [29] and pROC v1.15.0 [30] R packages. 

Accuracy is a measure of the overall correctness of the model’s classification:(1)$$\mathrm{Accuracy}=\frac{TP+TN}{TP+FP+FN+TN}$$

Sensitivity, or true positive rate (TPR), measures how often the model predicts that a sequence is resistant to a drug when it is actually resistant:(2)$$\mathrm{Sensitivity}=\frac{TP}{TP+FN}$$

Specificity, or true negative rate (TNR), measures how often the model predicts that the sequence is not resistant to a drug when it is actually not resistant:(3)$$\mathrm{Specificity}=\frac{TN}{TN+FP}$$

The false positive rate (FPR) measures how often the model predicts that a sequence is resistant to a drug when it is actually not resistant:(4)$$\mathrm{FPR}=\frac{FP}{FP+TN}$$

The false negative rate (FNR) measures how often the model predicts that a sequence is not resistant to a drug when it is actually resistant:(5)$$\mathrm{FNR}=\frac{FN}{FN+TP}$$

The F1 score measures the harmonic mean of precision of recall and is often preferred to accuracy when the data has imbalanced classes:(6)$$F1=2\cdot \frac{precision\cdot recall}{precision+recall}$$

Binary cross-entropy is a measure of the error of the model for the given classification problem, which is minimized by the neural network during the training phase. Thus, this metric represents performance during training but not necessarily performance on the testing dataset, which is used for evaluation. Binary cross-entropy is calculated as follows, where *y* is a binary indicator of whether a class label for an observation is correct and *p* is the predicted probability that the observation is of that class:(7)cross-entropy=−(ylog(p)+(1−y)log(1−p))

AUC measures the area under the receiving operator characteristic (ROC) curve, which plots true positive rate against false positive rate. AUC is also commonly used in situations where the data has imbalanced classes, as the ROC measures performance over many different scenarios.

### 2.4. Model Interpretation

Model interpretation analysis was conducted in R/RStudio using the permutation feature importance function implemented in the IML package v0.9.0 [23]. This function is an implementation of the model reliance measure [31], which is model-agnostic. Put simply, permutation feature importance is a metric of change in model performance when all data for a given feature is shuffled (permuted) and is measured in terms of 1-AUC. Feature importance plots were rendered using the ggplot2 package and annotated with known DRM positions using the Stanford database [9], both for the top 20 most important features and across the entire gene region.

### 2.5. Phylogenetics

In addition to deep learning-based analysis, we reconstructed phylogenetic trees for all datasets in order to empirically test whether resistant and non-resistant sequences formed distinct clades and to visualize evolutionary relationships present in the data. ModelTest-NG v0.1.5 [32] was used to estimate best-fit amino acid substitution models for each dataset for use in phylogeny reconstruction. The selected models included HIVB (FPV, ATV, TPV, and all PI), FLU (IDV, LPV, SQV, and DRV)–which has been shown to be highly correlated with HIVb [33], JTT (NFV, ETR, RPV, 3TC, D4T, DDI, TDF, and all NRTI), and JTT-DCMUT (EFV, NVP, ABC, AZT, and all NNRTI). We then used RAxML v8.2.12 [34] to estimate phylogenies for each data set using the maximum likelihood optimality criterion and included bootstrap analysis with 100 replicates to evaluate branch support. Both ModelTest-NG and RAxML were run within the CIPRES Web Interface v3.3 [35]. Trees were then annotated with drug resistance classes using iTOL v4 [36]. The approximately unbiased (AU) test for constrained trees [37] was used to test the hypothesis that all trees were perfectly clustered by drug resistance class using IQ-Tree v1.6.10 [38], with midpoint rooting used for all trees.

## 3. Results

### 3.1. Classifier Performance

Here, we compared the performance of three deep learning architectures for binary classification of HIV sequences by drug resistance: multilayer perceptron (MLP), bidirectional recurrent neural network (BRNN), and convolutional neural network (CNN) (Table 2). The reported metrics are averages taken from 5-fold cross-validation. Average accuracy across folds ranged from 65.9% to 94.6% for the MLPs, from 72.9% to 94.6% for the BRNNs, and 86.2% to 95.9% for the CNNs (Table A1, Table A2 and Table A3). Due to the noted class imbalances in the data, accuracy is not an ideal metric to compare performance, so we additionally considered AUC and the F1 score, both of which are more appropriate in this case. Average AUC across folds ranged from 0.760 to 0.935 for the MLPs, from 0.682 to 0.988 for the BRNNs, and 0.842 to 0.987 for the CNN models (Table A4; Figure 2, Figure 3 and Figure 4). Average F1 score across folds ranged from 0.224 to 0.861 for the MLPs, from 0.362 to 0.944 for the BRNNs, and 0.559 to 0.950 for the CNNs (Table A4). Across all models and all three overall performance metrics (accuracy, AUC, and F1), average performance was best for PI datasets, followed by NRTI and then NNRTI (Table 2). All three overall performance metrics also indicate that the CNN model showed the best performance of the three. False negative rates were similar among BRNNs and CNNs, both of which were notably lower than that of MLPs. Average false positive rate was notably lower for the BRNN and CNN models than the MLP model, while false negative rate remained within a more consistent range.

### 3.2. Model Interpretation

On average in the MLPs, of the 20 most important features returned by permutation feature importance analysis, 2.875 PI features (range 1–5 features), 3.167 NRTI features (range 1–5), and 1.75 NNRTI features (range 1–3) correlated to known DRM positions. For the BRNNs, an average of 5 PI features (range 2–7), 3 NRTI features (range 2–5), and 3 NNRTI features (range 1–6) correlated to known DRM positions. For the CNNs, an average of 5.75 PI features (range 3–7), 4.333 NRTI features (range 1–6), and 5.25 NNRTI features (range 5–6) correlated to known DRM positions. In total for MLPs, 50% (9/18) of datasets returned the most important feature being a known DRM position (PI: 0/8; NRTI: 5/6; NNRTI: 4/4), while BRNNs and CNNs fared better. A total of 88.9% (16/18) and 100% (18/18) of BRNNs and CNNs, respectively, returned a known DRM position as the most important feature position (BRNN PI: 7/8; NRTI: 6/6; NNRTI: 3/4; CNN PI: 8/8; NRTI: 6/6; NNRTI: 4/4) (Figure 5, Figure 6 and Figure 7, Appendix A). In the CNN classifiers for PI datasets, of features which were among the 20 most important features but are not known DRM-associated positions, four were identified in half (4/8) of the PI datasets. Similarly, for NNRTI datasets, five such features were identified in half (2/4) of the datasets and for NRTI datasets, three such features were identified in half (3/6) of the datasets. The majority of features with high importance but no known DRM association were identified in less than half of the datasets, across PI, NNRTI, and NRTI datasets. A majority of such features were only identified in one dataset for both NRTI and NNRTI, and in only one or two datasets in PI (Appendix A).

### 3.3. Phylogenetics

Phylogenetic trees were constructed by drug class (Figure 8) as well as for all 18 drug-specific datasets (Figure 9). For all datasets, the AU test for constrained trees rejected the null hypothesis of perfect clustering of resistant vs. nonresistant sequences (*p*-value range [2.12e-85,1.57e-4]). Visual clustering of resistant sequences is seen primarily in PI datasets, with some clustering apparent in NRTI datasets and little to none in NNRTI datasets. Yet few clades were supported by bootstrap replicate values, and so no trends about drug resistance prevalence within clades could be identified with statistical support.

## 4. Discussion

### 4.1. Model Performance

Here we compare performance metrics from cross-validation of the described multilayer perceptron (MLP), bidirectional recurrent neural network (BRNN), and convolutional neural network (CNN) models. Overall model performance metrics (accuracy, AUC, and F1 score) indicate that the CNN models were the best performing of those included in this study, which is consistent with their known success in the direct interpretation of sequencing data [21]. Thus, our results contribute to existing literature supporting CNN models as a useful architecture for sequencing data tasks, in addition to their widespread use in biomedical imaging. A further study exploring best practices for training CNN models on genomic sequencing data would be of great benefit to the development of computational tools in the HIV sphere and for other viral and human sequencing data. BRNN models also showed high performance in many cases, particularly for PI datasets, likely due to the importance of directionality in genetic sequence data. CNN models additionally displayed the lowest false positive rates, on average, for each drug class. These results suggest that, of the three considered, the CNN architecture would be the most suitable architecture for genotypic drug resistance tests, with BRNNs also being comparable in performance. 

The MLP models were included in this study as a baseline for comparison to the BRNN models and CNN models, and the notable increase in performance of the latter two confirm that more complex models are necessary for greater performance in this classification task. Further, it is notable in MLP models that none of the first most-important features from PI datasets were true DRM positions. It is possible that if variants are phylogenetically linked–as is suggested by the greater clustering of resistant sequences in these data–DRM-associated signals from these datasets are effectively non-independent, resulting in a reduction in statistical power [39]. This effect would likely be particularly troublesome for feed-forward models such as MLPs, whereas more advanced models such as BRNNs and CNNs could have greater success by identifying higher-level patterns such as those resulting from sequential ordering of features. Additional phylogenetic analysis would be necessary to address this hypothesis. 

While CNN performance metrics were quite high across all datasets–with the lowest average AUC being 84% (Table A4)–it should be noted that class imbalances may have contributed to these high numbers, though AUC is more robust to class imbalance than other measures (i.e., accuracy). Overall, these results are informative for the choice of deep learning architectures in future drug resistance prediction tool-development involving deep learning models. 

### 4.2. Model Interpretation

Based on permutation feature importance analysis, CNN models displayed the highest average number of known drug resistance mutation (DRM) loci within the top 20 most important features and were the only architecture for which the most important feature was a known DRM position in all 18 drug datasets (18/18). In contrast, only half of MLP models (9/18) identified a DRM position as the most important feature, while this was the case for the majority of the BRNN models (16/18). These results, combined with those of overall model performance, suggest the use of CNN models in deep learning-based drug resistance testing tools. Further, the identification of DRM positions among the most important features across datasets and architectures suggest that classification was largely based on drug resistance mutations, as opposed to artifacts or confounding factors (e.g., subtype, non-resistance mutations). Additionally, datasets for which the deep learning models had greater success in identifying known DRM loci as important features displayed higher performance, regardless of architecture, indicating a potential link between neural network functionality and evolutionary signals such as positive selection. 

All models highly weighted several features which do not correspond to known DRM positions. These positions could be further explored as potential DRMs through in vitro studies in future work. It is possible that these sites could be linked to known DRMs, although it is thought that linkage in HIV-1 is limited to a 100 bp range as a result of frequent recombination [40]. All but one of the features identified in at least half of the datasets for a drug class occurred within 100 bp (33 amino acid) range of a known DRM position, suggesting that linkage could be present. This observation is limited, though, by the availability of only consensus sequences–and not haplotype data–for these samples, as variants in the consensus may not appear together on any one haplotype [41,42]. It is also known that epistasis is a key factor to the evolution of HIV-1 drug resistance and that entrenchment of DRMs occurs in both protease and reverse transcriptase [43], which may also contribute to the importance of these features. The observation that a majority of highly ranked features which do not correlate to known DRMs were identified in only one or two datasets points to potential bias in the training data, which could have resulted from variation in HIV subtype or resistance to one or more additional ART drugs.

### 4.3. Phylogenetics

Our initial, null hypothesis was that resistant and non-resistant sequences would cluster among themselves due to similarities at DRM loci. However, the AU test rejected all constrained trees with high confidence, indicating that resistant and non-resistant sequences do not entirely separate in any of the phylogenies. While some trends in the clustering of drug resistant sequences in the phylogenies are visibly apparent, low bootstrap support values precluded statistical analysis of clustering in the context of model performance. Low bootstrap support values are known to be typical in HIV-1 phylogenies due to high similarity among sequences and potential recombination events [44]. This observation demonstrates that phylogenetics alone is not sufficient to determine drug resistance status, further necessitating the development and use of computational tools designed for drug resistance prediction for clinical and public health applications.

### 4.4. Limitations

The primary limitations of this study, and of many others which apply machine learning to biomedical data more broadly [45], are due to limited data availability. Due to sequencing costs and privacy concerns, the public availability of large quantities of HIV sequence data along with clinical outcomes such as drug resistance is inherently difficult. In addition, data that are available regarding HIV drug resistance exhibit notable class imbalance between resistant and non-resistant sequences, as shown here (Table 1), another noted challenge in deep learning for genomics [46]. In this study, these limitations were addressed using methods including stratified cross-validation and cost-sensitive learning; however, dataset size remained a limiting factor for performance in several cases. 

Additionally, it is plausible that dependencies exist between resistance phenotypes for different drugs (e.g., resistance to one PI drug may confer resistance to another PI drug), which could be highly informative. This is especially relevant as combination therapy is standard in HIV care [47]. These dependencies were not addressed in this study as all categories of drug-specific phenotype data were not available for all of the sequences. Furthermore, the availability of only amino acid data–and not the corresponding nucleotide sequences–limited the extent of evolutionary analysis possible, as most established methods rely on nucleotide data. Lastly, the lack of subtype information limits the generalization of these results to the global HIV epidemic.

### 4.5. Future Work: Further Applications of Interpretable Deep Learning in HIV

Here, we present an initial attempt to apply interpretable deep learning techniques to classify amino acid sequences by drug resistance phenotypes. We recognize that several challenges with integrating multi-omics data and machine learning are currently present [21,22,45,46,48], and so we have detailed several extensions of this work that would be beneficial to the field as more and better data become publicly available and as methods evolve to adapt to big-data difficulties. We intend for this study to serve as a proof of concept for incorporating model interpretation with biological data for novel analysis of viral genomics data. 

#### 4.5.1. Intra-Patient and Temporal Data

This work demonstrates one application of model interpretation to deep learning-based HIV drug resistance classification. A similar framework could be quite useful in studying other aspects of HIV drug resistance via sequence data. All sequences in the Stanford database are consensus sequences, which typically do not resemble any one haplotype but, rather, an ancestral sequence of the haplotypes present in the viral population [41]. Due to the high intra-patient variability of HIV, it is plausible that novel insights about drug resistance could be made from applying a similar approach to haplotypes obtained from NGS [49,50]. A large-scale dataset including both haplotype data and clinical phenotypes would be necessary to achieve this aim. Moreover, evolutionary data is inherently dependent on time scales. As time goes on, haplotype frequencies will be variable, and so multiple sampling points would be necessary for a robust analysis of drug resistance evolution in intra-patient data.

#### 4.5.2. Multi-Omics Approaches

Here, our approach has focused largely on a population genetics framework, which has been previously applied to HIV genomics data successfully [40]. Due to limitations of working with amino acid sequences, we have been constrained in population genetic analysis. Notably, nucleotide data would facilitate selection analysis (dN/dS) and better characterized co-evolutionary interactions [51,52,53]. In recent years, however, it has become increasingly evident that genomics–or any other single data type–is insufficient to fully capture the etiology and progression of a disease [48]. Numerous studies have identified the role of virus-host interactions and other gene expression pathways in HIV-1 drug resistance [54,55,56], pointing to the need for a genomics-transcriptomics database on this subject. Furthermore, integrating proteomics data such as tertiary structure, hydrogen bonding, and amino acid polarity and composition would help to expand our knowledge of the role amino acids and the accumulation of amino acid changes–together and individually–play in the development of drug interaction and resistance pathways [57,58].

#### 4.5.3. Applications to Other Viruses

Drug resistance in treatment-naive patients coinfected with HIV and HCV is not well studied [59]. Yet, HIV/HCV coinfection has been associated with selection patterns at DRM loci distinct from those of mono-infected HIV patients [60], and so data from both viruses may be informative for drug resistance patterns in coinfected patients. In future work, an interpretable deep learning approach could be used to identify drug resistance patterns across both viral genomes in order to account for all possible evolutionary pathways for drug resistance. Moreover, drug resistance develops in other viruses, and thus this approach could be applied to other viruses of interest. Particularly, the recent SARS-CoV-2 pandemic has instigated the use of drug repurposing [61], highlighting the importance of characterizing antiviral drug resistance in HIV-1 patients so that this knowledge can be readily adapted to other infectious agents for which previously tested ART drugs may have clinical utility.

## 5. Conclusions

This study benchmarks the performance of deep learning methods for drug resistance classification in HIV-1. While such classification methods have been implemented previously, this study further compares the performance of several ML architectures and, most notably, addresses model interpretability and its biological implications. Identification of known DRM positions within the neural network, as revealed by feature importance analysis, showed correspondence to higher classification performance. Furthermore, this link suggests the investigation of other important features as biologically relevant loci. This work demonstrates the utility of interpretable machine learning in studying HIV drug resistance and develops a framework which has many important applications in viral genomics more broadly.

## Figures and Tables

**Figure 1 viruses-12-00560-f001:**
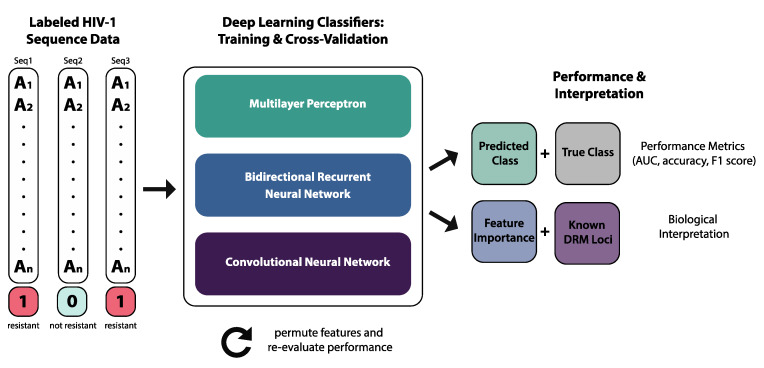
Overview of deep learning methods.

**Figure 2 viruses-12-00560-f002:**
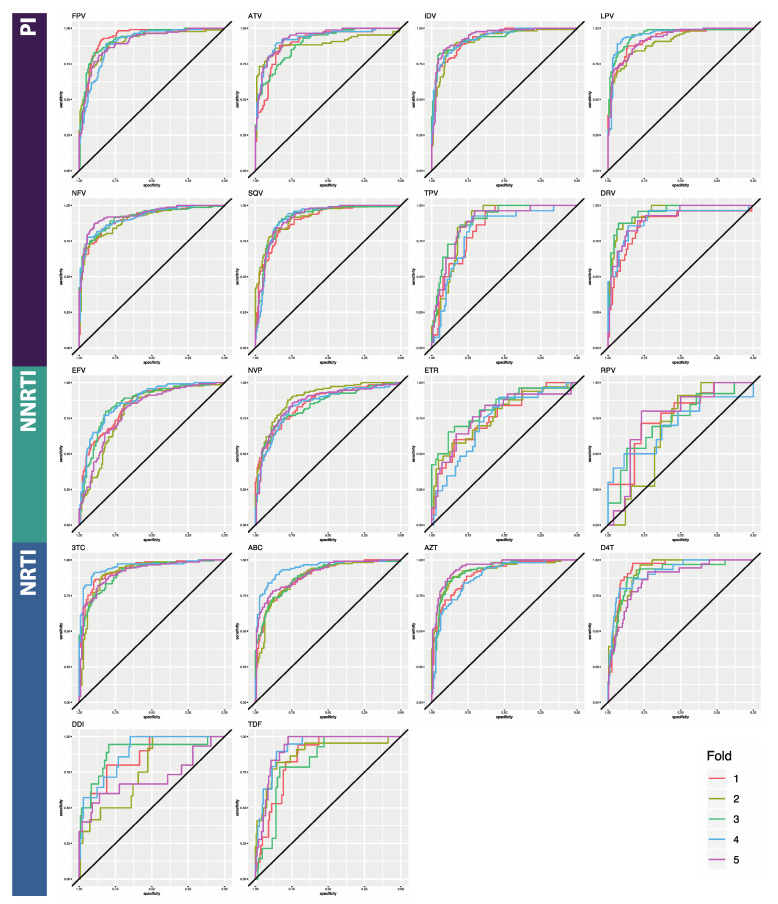
Receiving operator characteristic (ROC) curves for the Multilayer Perceptron classifier. Horizontal axes are specificity; vertical axes are sensitivity. The five curves represent the results of each cross-validation step. Curves which are closer to a ninety-degree angle in the top left corner represent better performance. Area under the curves is given in Table A4. Abbreviations: FPV = fosamprenavir; ATV = atazanavir; IDV = indinavir; LPV = lopinavir; NFV = nelfinavir; SQV = saquinavir; TPV = tipranavir; DRV = darunavir; 3TC = lamivudine; ABC = abacavir; AZT = azidothymidine; D4T = stavudine; DDI = didanosine; TDF = tenofovir disoproxil fumarate; EFV = efavirenz; NVP = nevirapine; ETR = etravirine; RPV = rilpivirine; PI = protease inhibitor; NRTI = nucleotide reverse transcriptase inhibitor; NNRTI = non-nucleotide reverse transcriptase inhibitor.

**Figure 3 viruses-12-00560-f003:**
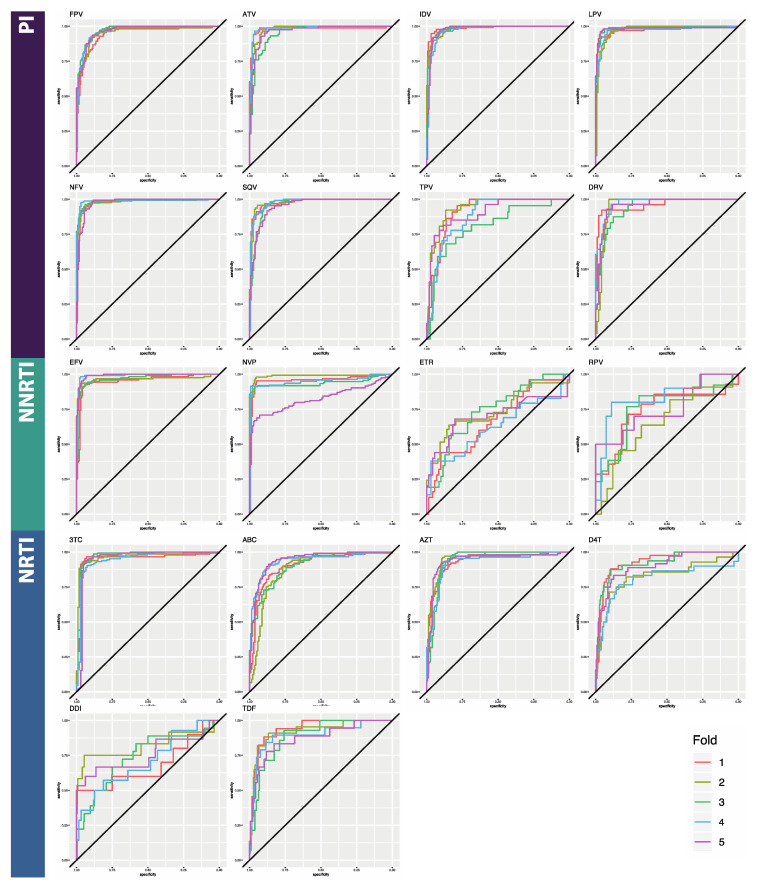
ROC curves for the Bidirectional Recurrent Neural Network classifier. Horizontal axes are specificity; vertical axes are sensitivity. The five curves represent the results of each cross-validation step. Curves which are closer to a ninety-degree angle in the top left corner represent better performance. Area under the curves is given in Table A4. Abbreviations: FPV = fosamprenavir; ATV = atazanavir; IDV = indinavir; LPV = lopinavir; NFV = nelfinavir; SQV = saquinavir; TPV = tipranavir; DRV = darunavir; 3TC = lamivudine; ABC = abacavir; AZT = azidothymidine; D4T = stavudine; DDI = didanosine; TDF = tenofovir disoproxil fumarate; EFV = efavirenz; NVP = nevirapine; ETR = etravirine; RPV = rilpivirine; PI = protease inhibitor; NRTI = nucleotide reverse transcriptase inhibitor; NNRTI = non-nucleotide reverse transcriptase inhibitor.

**Figure 4 viruses-12-00560-f004:**
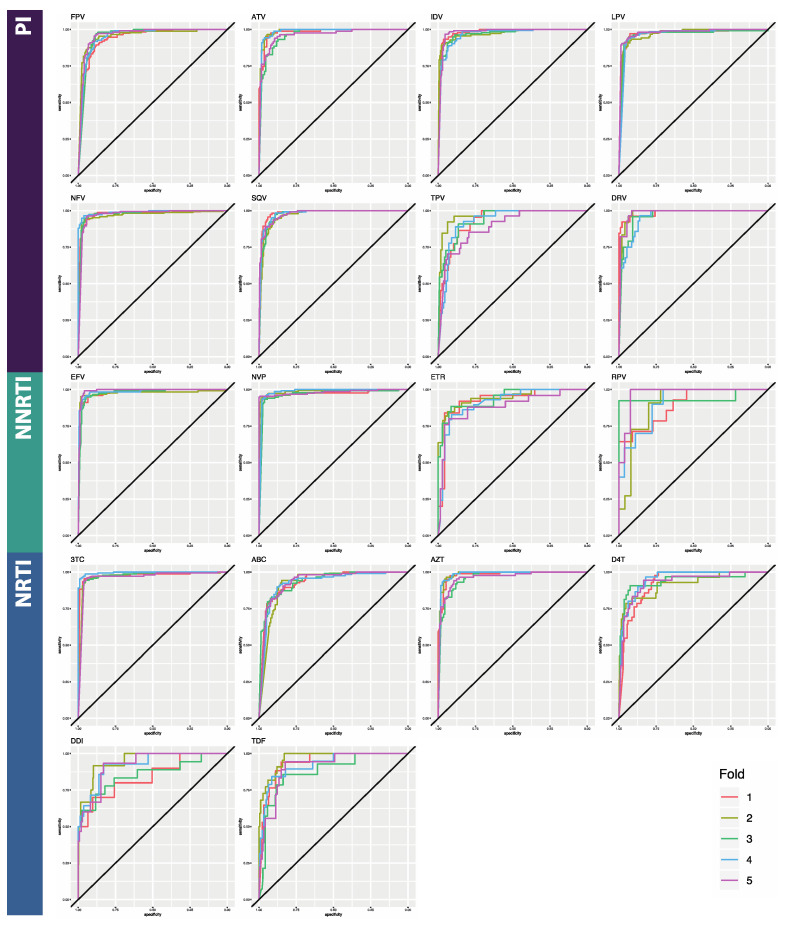
ROC curves for the Convolutional Neural Network classifier. Horizontal axes are specificity; vertical axes are sensitivity. The five curves represent the results of each cross-validation step. Curves which are closer to a ninety-degree angle in the top left corner represent better performance. Area under the curves is given in Table A4. Abbreviations: FPV = fosamprenavir; ATV = atazanavir; IDV = indinavir; LPV = lopinavir; NFV = nelfinavir; SQV = saquinavir; TPV = tipranavir; DRV = darunavir; 3TC = lamivudine; ABC = abacavir; AZT = azidothymidine; D4T = stavudine; DDI = didanosine; TDF = tenofovir disoproxil fumarate; EFV = efavirenz; NVP = nevirapine; ETR = etravirine; RPV = rilpivirine; PI = protease inhibitor; NRTI = nucleotide reverse transcriptase inhibitor; NNRTI = non-nucleotide reverse transcriptase inhibitor.

**Figure 5 viruses-12-00560-f005:**
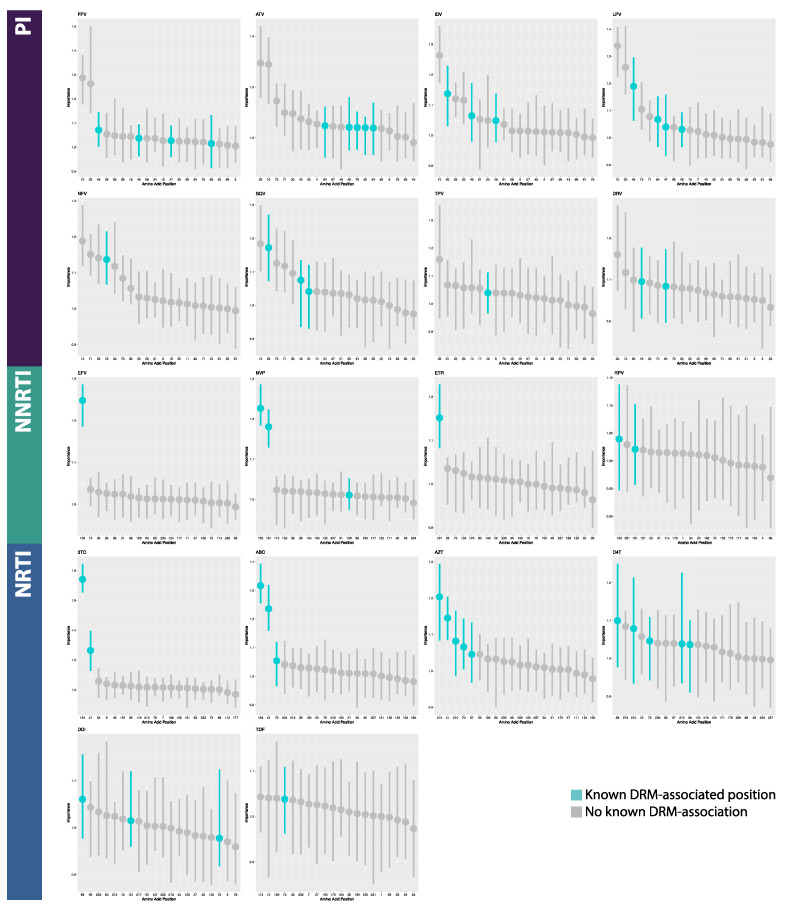
Annotated feature importance plots for the 20 most important features in Multilayer Perceptron classifiers. Horizontal axes are amino acid positions; vertical axes are feature importance (measured as change in 1-AUC). Abbreviations: FPV = fosamprenavir; ATV = atazanavir; IDV = indinavir; LPV = lopinavir; NFV = nelfinavir; SQV = saquinavir; TPV = tipranavir; DRV = darunavir; 3TC = lamivudine; ABC = abacavir; AZT = azidothymidine; D4T = stavudine; DDI = didanosine; TDF = tenofovir disoproxil fumarate; EFV = efavirenz; NVP = nevirapine; ETR = etravirine; RPV = rilpivirine; PI = protease inhibitor; NRTI = nucleotide reverse transcriptase inhibitor; NNRTI = non-nucleotide reverse transcriptase inhibitor.

**Figure 6 viruses-12-00560-f006:**
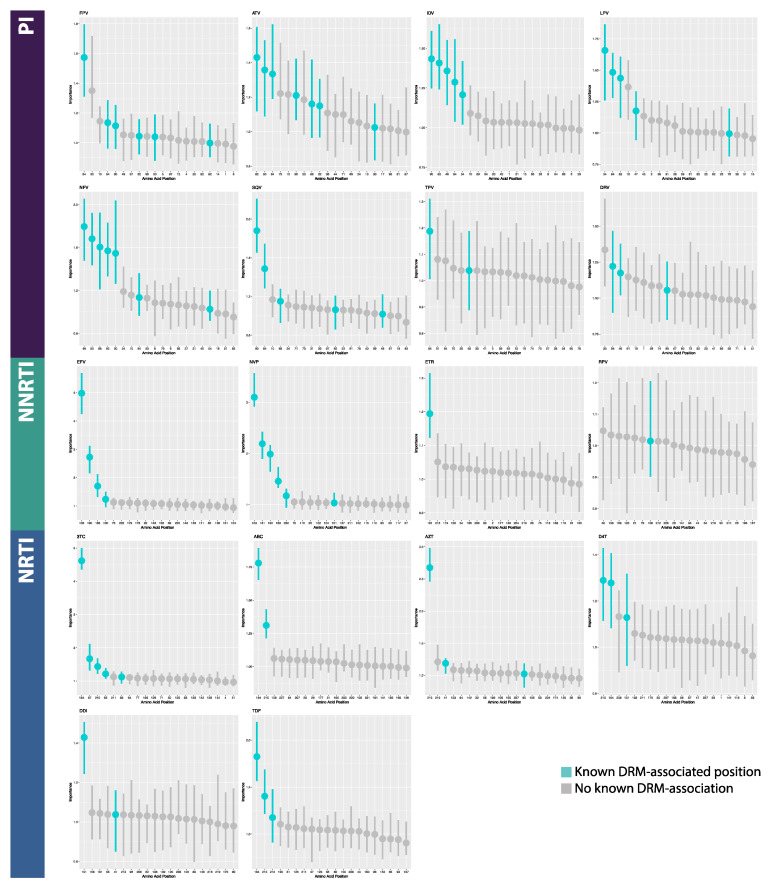
Annotated feature importance plots for the 20 most important features in Bidirectional Recurrent Neural Network classifiers. Horizontal axes are amino acid positions; vertical axes are feature importance (measured as change in 1-AUC). Abbreviations: FPV = fosamprenavir; ATV = atazanavir; IDV = indinavir; LPV = lopinavir; NFV = nelfinavir; SQV = saquinavir; TPV = tipranavir; DRV = darunavir; 3TC = lamivudine; ABC = abacavir; AZT = azidothymidine; D4T = stavudine; DDI = didanosine; TDF = tenofovir disoproxil fumarate; EFV = efavirenz; NVP = nevirapine; ETR = etravirine; RPV = rilpivirine; PI = protease inhibitor; NRTI = nucleotide reverse transcriptase inhibitor; NNRTI = non-nucleotide reverse transcriptase inhibitor.

**Figure 7 viruses-12-00560-f007:**
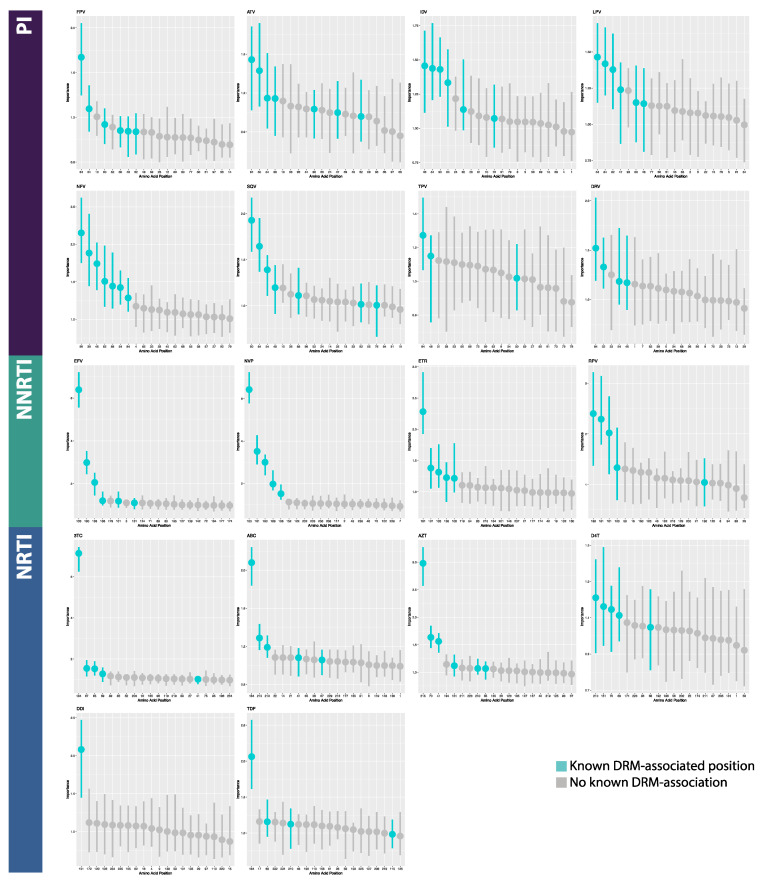
Annotated feature importance plots for the 20 most important features in Convolutional Neural Network classifiers. Horizontal axes are amino acid positions; vertical axes are feature importance (measured as change in 1-AUC). Abbreviations: FPV = fosamprenavir; ATV = atazanavir; IDV = indinavir; LPV = lopinavir; NFV = nelfinavir; SQV = saquinavir; TPV = tipranavir; DRV = darunavir; 3TC = lamivudine; ABC = abacavir; AZT = azidothymidine; D4T = stavudine; DDI = didanosine; TDF = tenofovir disoproxil fumarate; EFV = efavirenz; NVP = nevirapine; ETR = etravirine; RPV = rilpivirine; PI = protease inhibitor; NRTI = nucleotide reverse transcriptase inhibitor; NNRTI = non-nucleotide reverse transcriptase inhibitor.

**Figure 8 viruses-12-00560-f008:**
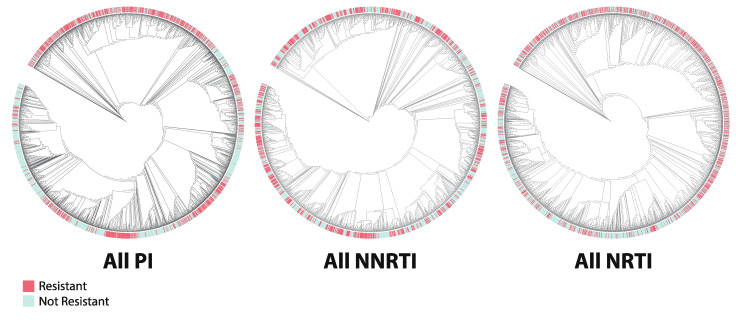
Annotated phylogenetic trees for each drug class. Resistant sequences are denoted by red labels; non-resistant sequences are denoted by blue labels. Abbreviations: FPV = fosamprenavir; ATV = atazanavir; IDV = indinavir; LPV = lopinavir; NFV = nelfinavir; SQV = saquinavir; TPV = tipranavir; DRV = darunavir; 3TC = lamivudine; ABC = abacavir; AZT = azidothymidine; D4T = stavudine; DDI = didanosine; TDF = tenofovir disoproxil fumarate; EFV = efavirenz; NVP = nevirapine; ETR = etravirine; RPV = rilpivirine; PI = protease inhibitor; NRTI = nucleotide reverse transcriptase inhibitor; NNRTI = non-nucleotide reverse transcriptase inhibitor.

**Figure 9 viruses-12-00560-f009:**
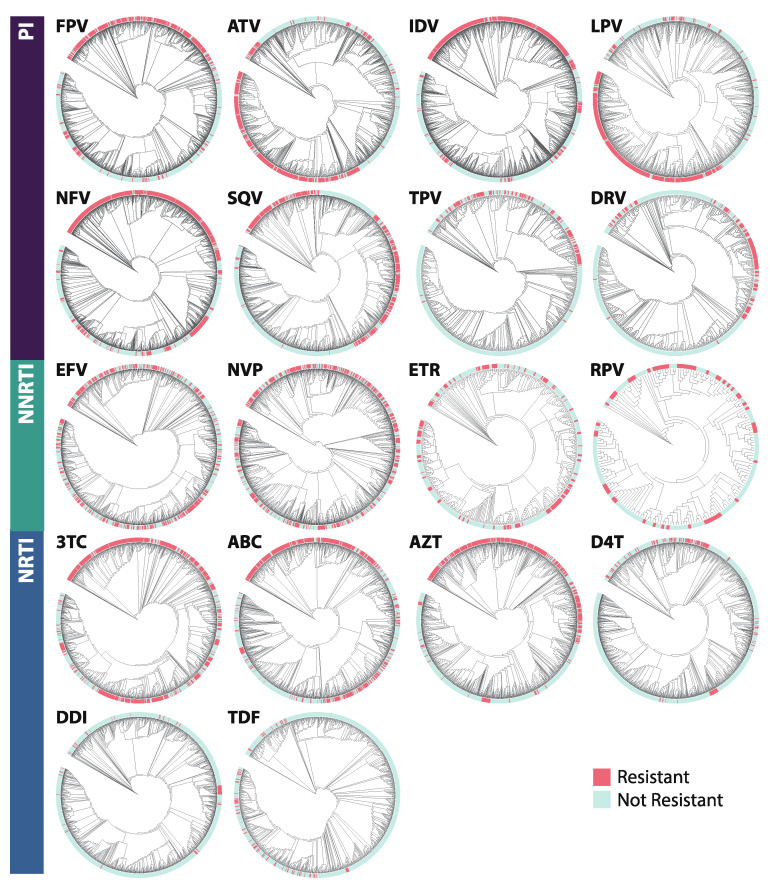
Annotated phylogenetic trees for each dataset. Resistant sequences are denoted by red labels; non-resistant sequences are denoted by blue labels. Abbreviations: FPV = fosamprenavir; ATV = atazanavir; IDV = indinavir; LPV = lopinavir; NFV = nelfinavir; SQV = saquinavir; TPV = tipranavir; DRV = darunavir; 3TC = lamivudine; ABC = abacavir; AZT = azidothymidine; D4T = stavudine; DDI = didanosine; TDF = tenofovir disoproxil fumarate; EFV = efavirenz; NVP = nevirapine; ETR = etravirine; RPV = rilpivirine; PI = protease inhibitor; NRTI = nucleotide reverse transcriptase inhibitor; NNRTI = non-nucleotide reverse transcriptase inhibitor.

**Table 1 viruses-12-00560-t001:** Description of datasets.

Drug	Drug Class	No. Sequences	Sequence Length	(%) Resistant
FPV	PI	1444	99	36.5
ATV	PI	987	99	43.4
IDV	PI	1491	99	43.7
LPV	PI	1267	99	44.7
NFV	PI	1532	99	52.8
SQV	PI	1483	99	37.6
TPV	PI	696	99	17.8
DRV	PI	605	99	21.5
All PI		2112	99	41
EFV	NNRTI	1471	240	41.5
NVP	NNRTI	1478	240	47.8
ETR	NNRTI	481	240	28.7
RPV	NNRTI	181	240	32
All NNRTI		1772	240	42
3TC	NRTI	1270	240	58.3
ABC	NRTI	1293	240	46.1
AZT	NRTI	1283	240	41.8
D4T	NRTI	1291	240	13
DDI	NRTI	1291	240	5.3
TDF	NRTI	1025	240	8.8
All NRTI		2129	240	40.4

Abbreviations: FPV = fosamprenavir; ATV = atazanavir; IDV = indinavir; LPV = lopinavir; NFV = nelfinavir; SQV = saquinavir; TPV = tipranavir; DRV = darunavir; 3TC = lamivudine; ABC = abacavir; AZT = azidothymidine; D4T = stavudine; DDI = didanosine; TDF = tenofovir disoproxil fumarate; EFV = efavirenz; NVP = nevirapine; ETR = etravirine; RPV = rilpivirine; PI = protease inhibitor; NRTI = nucleotide reverse transcriptase inhibitor; NNRTI = non-nucleotide reverse transcriptase inhibitor.

**Table 2 viruses-12-00560-t002:** Average performance metrics by drug class; standard deviation in brackets.

MLP								
*Class*	*Accuracy*	*Loss*	*TPR*	*TNR*	*FPR*	*FNR*	*AUC*	*F1*
PI	0.826 [0.039]	0.406 [0.058]	0.813 [0.041]	0.834 [0.061]	0.166 [0.061]	0.187 [0.041]	0.9 [0.009]	0.732 [0.135]
NNRTI	0.706 [0.095]	0.586 [0.143]	0.662 [0.140]	0.718 [0.105]	0.282 [0.105]	0.338 [0.140]	0.805 [0.006]	0.596 [0.158]
NRTI	0.795 [0.061]	0.470 [0.150]	0.725 [0.112]	0.807 [0.081]	0.193 [0.081]	0.275 [0.112]	0.895 [0.024]	0.593 [0.236]
**BRNN**								
*Class*	*Accuracy*	*Loss*	*TPR*	*TNR*	*FPR*	*FNR*	*AUC*	*F1*
PI	0.912 [0.032]	0.290 [0.087]	0.890 [0.065]	0.919 [0.028]	0.081 [0.028]	0.110 [0.065]	0.965 [0.014]	0.860 [0.106]
NNRTI	0.884 [0.104]	0.334 [0.247]	0.829 [0.165]	0.914 [0.088]	0.086 [0.088]	0.171 [0.165]	0.83 [0.021]	0.836 [0.180]
NRTI	0.896 [0.030]	0.357 [0.094]	0.751 [0.168]	0.910 [0.029]	0.090 [0.029]	0.249 [0.168]	0.89 [0.02]	0.698 [0.220]
**CNN**								
*Class*	*Accuracy*	*Loss*	*TPR*	*TNR*	*FPR*	*FNR*	*AUC*	*F1*
PI	0.919 [0.019]	0.745 [0.204]	0.865 [0.081]	0.939 [0.013]	0.061 [0.013]	0.135 [0.081]	0.968 [0.007]	0.871 [0.088]
NNRTI	0.911 [0.048]	0.614 [0.219]	0.845 [0.099]	0.945 [0.029]	0.055 [0.029]	0.155 [0.099]	0.945 [0.012]	0.861 [0.097]
NRTI	0.920 [0.035]	0.805 [0.277]	0.743 [0.182]	0.940 [0.040]	0.060 [0.040]	0.257 [0.182]	0.927 [0.016]	0.757 [0.163]

Abbreviations: FPV = fosamprenavir; ATV = atazanavir; IDV = indinavir; LPV = lopinavir; NFV = nelfinavir; SQV = saquinavir; TPV = tipranavir; DRV = darunavir; 3TC = lamivudine; ABC = abacavir; AZT = azidothymidine; D4T = stavudine; DDI = didanosine; TDF = tenofovir disoproxil fumarate; EFV = efavirenz; NVP = nevirapine; ETR = etravirine; RPV = rilpivirine; PI = protease inhibitor; NRTI = nucleotide reverse transcriptase inhibitor; NNRTI = non-nucleotide reverse transcriptase inhibitor.

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
