# Peer review of "Drug Resistance Prediction Using Deep Learning Techniques on HIV-1 Sequence Data"

_viruses, 2020, doi:10.3390/v12050560_

Round 1
Reviewer 1 Report
Following comments may be addressed before publication;
- Cited reference data set (Stanford HIV database) includes sequences from Integrase. Address why integrase dataset is not used in your model testing. Integrase drugs are part of current standard of care.
- As a part of performance metrics Binary cross-entropy (loss) was presented in table 2. But not discussed. As I understand higher the value in ‘loss’ indicate bad prediction. If this is true, line 223 and 224 for CNN model contradicts.
- Figure axis not readable for figures 5,6,7 include quality images prior to publication.
- Line 459 – “As both infections are treated using same drugs….” This statement is not true. Therapies for HIV and HCV are different. Re-phrase this line.
- Line 457 to 466 is not relevant to this paper. Consider deleting it or move to introduction as short summary.
Reviewer 2 Report
The manuscript is well-written. The authors utilized published HIV-1 Pol sequences with known mutations and corresponding drug-resistance, followed by using various “Deep learning Techniques” to study/identify drug resistance mutations and compare it to the actual phenotype data from PhenoSense assay.
Can the authors explain how their “CNN” model would help real-world HIV-1 patients? Can the authors predict novel HIV-1 Pol mutations that are drug-resistant and provide in-vitro data to support such predictions? Can the authors explain why all the models performed poorly in predicting drug-resistance from several drugs such as FPR and FNR? And ways to improve prediction against FPR and FNR?
Minor concerns:
1) Which subtype of “Wild-type HIV-1” was used in the PhenoSense assay? There are multiple subtypes among HIV-1 and these subtypes could have different sensitivity towards drugs. Can the authors comments on the effect of HIV-1 subtypes on their drug-resistance predictions?
2) Did the authors performed the PhenoSense assay in this study or the results of PhenoSense assay was obtained from other published papers? This part is not clear. The current manuscript sounds like the authors performed PhenoSense assay on all the sequences they tested.
